# Analyzing Citramalic Acid Enantiomers in Apples and Commercial Fruit Juice by Liquid Chromatography–Tandem Mass Spectrometry with Pre-Column Derivatization

**DOI:** 10.3390/molecules28041556

**Published:** 2023-02-06

**Authors:** Maho Umino, Mayu Onozato, Tatsuya Sakamoto, Mikoto Koishi, Takeshi Fukushima

**Affiliations:** Department of Analytical Chemistry, Faculty of Pharmaceutical Sciences, Toho University, 2-2-1 Miyama, Funabashi-shi 274-8510, Japan

**Keywords:** citramalic acid, high-performance liquid chromatography–tandem mass spectrometry (LC–MS/MS), benzyl 5-(2-aminoethyl)-3-methyl-4-oxoimidazolidine-1-carboxylate (CIM-C_2_-NH_2_)

## Abstract

Optically active citramalic acid (CMA) is naturally present as an acidic taste component in fruits, such as apples. The absolute configuration of CMA in such fruits was investigated by high-performance liquid chromatography–tandem mass spectrometry (LC–MS/MS) following pre-column derivatization with a chiral reagent, benzyl 5-(2-aminoethyl)-3-methyl-4-oxoimidazolidine-1-carboxylate. The developed LC–MS/MS method successfully separated the enantiomers of CMA using an octadecylsilica column with a resolution and separation factor of 2.19 and 1.09, respectively. Consequently, the *R*-form of CMA was detected in the peel and fruit of three kinds of apple at concentrations in the 1.24–37.8 and 0.138–1.033 mg/wet 100 g ranges, respectively. In addition, *R*- CMA was present in commercial apple juice, whereas no quantity was detected in commercial blueberry, perilla, or Japanese apricot juice.

## 1. Introduction

Citrus and other fruits contain organic acids that contribute to their sour taste. Dicarboxylic acids, such as malic acid (Mal), tartaric acid (Tar), and citramalic acid (CMA) are representative organic acids in such fruits, with the structure of CMA (2-methylmalic acid) being similar to that of Mal. CMA exists as a pair of optical isomers due to the presence of an asymmetric carbon at its 2-position, with the (−)-form (i.e., *R*-form) previously reported to be present in apple peels [1]. *R*-(−)-CMA is enzymatically produced by CMA synthase in plants and yeast from achiral pyruvic acid and acetyl coenzyme A (Appendix A) [2,3] through a metabolic pathway referred to as the “CMA pathway”.

*Aspergillus niger*, a filamentous fungus, and *Alcaligenes Xylosoxidans* IL142 bacteria have been recently reported to produce CMA from 3*S*-citramalyl CoA via itaconic acid [4,5]. The bacterial CMA-production pathway (Appendix A) has been suggested to give the *S*-form of CMA [4,5,6].

We recently developed a high-performance liquid chromatography–tandem mass spectrometry (LC–MS/MS) technique that involves forming a mixture of diastereomers through pre-column derivatization with benzyl 5-(2-aminoethyl)-3-methyl-4-oxoimidazolidine-1-carboxylate (CIM-C_2_-NH_2_) (Figure 1) [7]. Consequently, the enantiomers of CMA were separated by LC–MS/MS using an octadecylsilica (ODS) column, and we showed that *R*-CMA was present in a commercial red-wine sample [7]. To the best of our knowledge, to date, the presence of *R*-CMA in natural fruits has not been confirmed by chromatography. Therefore, to determine the absolute configuration of CMA in fruit samples, an LC–MS/MS technique capable of separating the enantiomers of CMA offering high efficiency is needed. Accordingly, in this study, we determined the contents and absolute configurations of CMA detected in three kinds of commercially purchased apples by LC–MS/MS. Furthermore, the presence of optically active CMA in commercial fruit juice was examined.

## 2. Materials and Methods

### 2.1. Chemicals

*R*- and *S*-CMA were purchased from Sigma (St. Louis, CA, USA) and Toronto Research Chemicals (Toronto, ON, Canada), respectively. Formic acid (HPLC-grade) was obtained from the Fujifilm Wako Pure Chemical Corporation (Osaka, Japan). Triphenylphosphine (TPP) and 2,2′-dipyridyl disulfide (DPDS) were obtained from Tokyo Chemical Industry Co., Ltd. (Tokyo, Japan). CH_3_CN (LCMS-grade) was obtained from Kanto Kagaku Co., Ltd. (Tokyo, Japan). Phosphate-buffered saline (PBS) was obtained from the Nissui Pharmaceutical Co., Ltd. (Tokyo, Japan). H_2_O was purified using a Milli-Q Labo system (Nihon Millipore Co. Ltd., Tokyo, Japan). *R*-CIM-C_2_-NH_2_ was synthesized following our previously reported method [7].

### 2.2. Optical Rotations

*R*- and *S*-CMA were dissolved in 1 M HCl to prepare 0.46 and 0.48 *w*/*v*% solutions, respectively. The optical rotation of each solution was measured with a P-2200 digital polarimeter (Jasco Corporation, Tokyo, Japan) using a 100-mm quartz cell.

### 2.3. Sample Preparation 

Commercially available *Toki*, *Tsugaru*, and *Sun-Tsugaru* apples were purchased from a local market in Tokyo. The apples were peeled manually using a kitchen knife, after which the peel and fruit were homogenized using a BM-FX08-GA kitchen mixer (Zojirushi Corporation, Osaka, Japan). For the analysis of fruit of apple, the fruit was homogenized, then centrifuged at 3500 rpm for 15 min, after which the supernatant was centrifuged at 12,000 rpm for 15 min at 4 °C. The resulting supernatant (10 μL) was subjected to analysis. For the peel of apple, water was added to the peel in a weight ratio of 2:1 and the mixture was homogenized. The homogenate was centrifuged at 3500 rpm for 15 min, after which the supernatant was centrifuged at 12,000 rpm for 15 min at 4 °C. The obtained supernatant (diluted 50 times with H_2_O, 10 µL) was derivatized by first mixing it with an internal standard (IS) (10 µM sodium d-lactate (^13^C_3_, 98%) and 1.0 mM l-lactate -3,3,3-*d*_3_ in PBS, 10 µL), TPP and DPDS in CH_3_CN (250 mM, 10 µL each), and then with *R*-CIM-C_2_-NH_2_ in CH_3_CN (10 mM, 10 µL). The resultant mixture was allowed to stand for 30 min at 60 °C. The derivatization reaction was terminated by adding 50 µL of HCO_2_H/H_2_O/CH_3_CN (0.05/80/20, *v/v/v*). 

### 2.4. LC–MS/MS

An LCMS-8040 (Shimadzu corporation, Kyoto, Japan) LC–MS/MS system fitted with an InertSustain C18 (150 mm × 2.1 mm i.d., 3 μm) column (Tokyo, Japan) was used, with the CTO-20A (Shimadzu) column oven maintained at 40 °C. The mobile phase consisted of 0.05% formic acid in H_2_O (A) and 0.05% formic acid in CH_3_CN (B) with the gradient elution performed at 0.3 mL/min using the following time program: 0–30 min: A 85%, 30.01–40 min: A 80%, 40.01–74 min: A 70%. 

The MS was operated in positive ion-mode with electrospray ionization as the ion source. The heat-block and desolvation-line temperatures were set at 300 and 500 °C, respectively. The nebulizer and drying gas flow rates were 3.0 and 15 L/min, respectively. The collision-induced dissociation (CID) gas (Argon) pressure and voltage of ion spray were 230 kPa and 5 kV, respectively. The MRM method of CMA was 667.1 > 91.05 and 362.2 (Appendix A), and the IS was 353.45 > 91.15, which were used in our previous paper [7]. The MRM chromatogram of the IS was shown in Appendix A.

### 2.5. Validation

#### 2.5.1. Calibration Curves

The linear calibration curves of fruits were prepared by plotting the peak area ratio against the concentration (5.0, 10, 25, 50, and 100 μM for fruit, and 100, 250, 500, 1000, and 2500 μM for peel). The standard solution was diluted 50 times, derivatized, and examined using the same method employed for the apple samples.

#### 2.5.2. Intra and Inter-Day Precisions

*Sun-Tsugaru* fruit or peel samples were derivatized as described in Section 2.3. Intra-day precision was determined four times (n = 4), whereas inter-day precision was determined over 4 d (n = 4).

#### 2.5.3. Recovery

The *Sun-Tsugaru* fruit or peel solution (10 μL) was added to the standard solution (fruit: 25 and 50 μM, peel: 500 and 1000 μM, 10 μL) and water (480 μL). These samples (10 μL, n = 4) were treated as described in Section 2.3, with recoveries determined as previously reported [8].

## 3. Results

### 3.1. R- and S-CMA Standards

The optical rotations of the *R*- and *S*-CMA standards obtained from the suppliers were measured first; *R*-CMA (0.46 in 1 M HCl) and *S*-CMA (0.48 in 1 M HCl) exhibited rotations of –11.4341 and +18.0296, respectively. 

The *R*- and *S*-CMA standards were derivatized into a pair of diastereomers using *R*-CIM-C_2_-NH_2_, a chiral derivatization reagent [7]. 

Figure 1a shows the MS/MS spectra of the *R*- and *S*-CMA derivatives; it is obvious that both spectra are similar. Therefore, the cleavage pattern of the derivative with CIM-C_2_-NH_2_ might not differ between the enantiomers (Figure 1b).

Figure 2 shows the time-course CMA-derivatization profiles with CIM-C_2_-NH_2_ in the presence of TPP and DPDS, which plateau after 30 min; consequently, the derivatization time was set as 30 min. Figure 3a shows the chromatograms of the *R*- and *S*-CMA standards following derivatization, revealing that the peak corresponding to *R*-CMA eluted at 66.5 min, whereas that corresponding to *S*-CMA eluted at 68.6 min, with the resolution and separation factor (separation parameters) determined to be 2.19 and 1.09, respectively. In addition, neither optically active CMA was observed to racemize (Figure 3a). Hence, the absolute configuration of CMA present in apple samples can be determined by the developed LC–MS/MS method; consequently, the absolute configurations of CMA present in various apple samples were investigated next.

### 3.2. CMA in Apples and Commercial Fruit Juice

To investigate the organic acid contents of natural apples, apple fruit and peel were homogenized separately and then diluted 50 times with H_2_O. The diluted extract was centrifuged and directly derivatized with the CIM-C_2_-NH_2_ pre-column reagent, in the presence of TPP and DPDS as condensing agents. Figure 3b–g show the representative chromatograms of natural apple samples, revealing that these samples contain only *R*-CMA. The small peaks shown in Figure 3c,e,g were considered as noise peaks originating from the derivatization reagent, CIM-C_2_-NH_2_, because these small peaks were also detected in the chromatogram of a blank sample, which is a sample prepared without apple peel or fruit. The amount present per 100 g of each sample is shown in Figure 4. 

Table 1 shows the *R*-CMA validation data determined for apple peel and fruit samples using the developed LC–MS/MS method. The relative standard deviations (RSDs) of peel samples are 1.33% and 1.72% for intra- and inter-day precisions, respectively, with 93% recoveries (each). The apple fruit samples exhibited intra- and inter-day precision RSDs of 1.84% and 7.89%, respectively, with recoveries of 87 and 86%. In addition, the limit of detection of the method (LOD) was determined to be 15.40 fmol (signal to noise ratio (S/N) = 3).

The *R*-CMA contents (mean ± SE) were 36.5 ± 1.21, 10.5 ± 2.12, and 8.74 ± 5.87 mg/wet 100 g in the peel (*n* = 3), whereas 0.922 ± 0.066, 0.522 ± 0.141, and 0.292 ± 0.114 mg/wet 100 g in the fruit (n = 3) of *Toki*, *Tsugaru*, and *Sun-Tsugaru*, respectively. As concluded from Figure 4, the *R*-CMA content of the different apple types varied significantly. In addition, commercial apple juice was found to contain considerable amounts of *R*-CMA, whereas no CMA was detected in blueberry, perilla, or Japanese apricot juice (Figure 5).

## 4. Discussion 

CMA is a useful industrial starting material for the synthesis of methacrylic acid, which is polymerized to form transparent and solid plastics [9]. Consequently, metabolically engineered *Escherichia Coli* bacteria that produce CMA have been intensively researched [10,11,12]. The CMA biosynthetic pathway (Appendix A) produces various compounds including essential branched-chain amino acids, such as isoleucine, valine, and leucine. Some straight- and branched-chain carboxylate esters are also produced, which contribute to the aroma of fruit. It is reported that the CMA biosynthetic pathway is fundamentally involved in the procedure of fruit ripening [3,13]. The first product of this pathway is *R*-(−)-CMA (Appendix A), whereas some bacteria produce the *S*-form of CMA (Appendix A) [4,5,6]. 

Useful information on the configuration of naturally occurring CMA has been reported recently, stating that two pathways are capable of enzymatically producing optically pure CMA enantiomers: *S*-CMA is formed in bacteria [4] and fruits [14], whereas *R*-CMA is produced in other fruits [1], yeast [15], and *Methanogenic Archaea* [16]. 

Chromatography is a useful method for determining the absolute configuration of CMA because it can provide enantiomeric ratios in an operationally simple manner. With regard to enantiomeric separation methodologies reported previously for CMA, some studies using a chiral stationary phase in HPLC have been reported [17,18]; however, such techniques have been rarely applied to real food samples. Indeed, the little information on the enantiomeric separation of CMA in various natural samples appears to exist, and the stereochemistry of CMA present in various kinds of plants, fruits, or bacterial samples remains unclear. With this in mind, we pre-column derivatized the two carboxylic acid groups of CMA with CIM-C_2_-NH_2_, a chiral reagent, which enabled the CMA enantiomers present in fruit samples to be analyzed using LC–MS/MS.

The present data on *Toki*, *Tsugaru*, and *Sun-Tsugaru* apples reveal that the fruits contain only *R*-CMA. CMA is biosynthesized enzymatically from non-chiral pyruvic acid and acetyl coenzyme-A by CMA synthase via the CMA pathway in plants [3,9], and CMA is subsequently converted to citraconic acid by 2-isopropylmalate isomerase (IPMI). Therefore, it is likely that *R*-CMA is biosynthesized by CMA synthase. In addition, IPMI is likely to recognize *R*-CMA and produce olefins through dehydration processes in apples (Appendix A). 

(−)-CMA (i.e., *R*-CMA) has been previously reported to accumulate in beer produced using respiratory-deficient beer yeast [15]. In addition, Howell et al. demonstrated that *R*-CMA is produced by a recombinant CMA synthase, which uses a *Methanogenic Archaea* bacterial *cimA* gene, by gas-chromatography (GC) using a chiral (G-TA Chiraldex) capillary GC column [16]. These reports support the notion that CMA synthase can produce *R*-CMA in microorganisms. We previously reported the presence of *R*-CMA in a commercial fermented product (i.e., wine) [7].

Meanwhile, only the *S*-form of Mal (i.e., l-Mal) has been reported to exist in fruits. Indeed, we observed only *S*-Mal in apples by column-switching HPLC using a fluorescence detection method [19]. Therefore, *S*-Mal is produced by fumarase (EC 4.2.1.2) in fruits via the Krebs cycle. Further research into other fruits and plants, along with biochemical and molecular-biology experiments, are necessary to elucidate the mechanisms responsible for the exclusive production of *R*-CMA in fruits and plants.

The present data also indicate that more *R*-CMA is present in the peels rather than the fruits of *Tsugaru*, *Sun-Tsugaru*, and *Toki* apples. These results are consistent with previous studies reporting that CMA is more abundant in peels than in fruits [1], although the CMA configuration had not been clarified. 

Noro et al. previously suggested that red apple peel contains higher amounts of CMA than yellow peel, and that CMA may contribute to the production of the red anthocyanin pigment in the skins of apples [20]. The present study revealed that the CMA content in *Toki*, whose peel is yellow, was higher than CMA content in *Tsugaru* or *Sun-Tsugaru*, whose peels are red. Therefore, the relationship between peel color and CMA content remains unclear, with Di Matteo et al. recently reporting no relationship between them [21]. 

To certify that apples contain only *R*-CMA, we examined two types of commercial apple juice manufactured using cultivated apples, as well as a selection of other juices. It was revealed that only apple juice contains *R*-CMA, whereas no CMA was detected in the other commercial juices, including blueberry, Japanese apricot, and perilla. 

With regard to the limitation of the present study, the enantiomeric form of CMA present in other fruits remains unclear. In addition, the effect of seasonal changes as well as ripening or climacteric differences on CMA chirality will be the subjects of future studies. 

The present study revealed that only *R*-CMA is present in apples, and that pre-column derivatization with CIM-C_2_-NH_2_ (a chiral reagent) together with LC–MS/MS is highly efficient for enantiomerically separating chiral acidic compounds, such as CMA, present in fruit samples.

## Data Availability

Research data are not shared.

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
