# Peer review of "Analyzing Citramalic Acid Enantiomers in Apples and Commercial Fruit Juice by Liquid Chromatography–Tandem Mass Spectrometry with Pre-Column Derivatization"

_molecules, 2023, doi:10.3390/molecules28041556_

Round 1
Reviewer 1 Report
The study by Umino et al. aimed to establish a quantitative method for optically active citramalic acid (CMA). After pre-column derivatization, LC-MS/MS was used to separate the enantiomers with reasonable accuracy and precision. Addressing the following comments should improve the quality of the manuscript:
1. The authors may consider rewriting this article as a shorter communication.
2. Only R-CMA was found in the samples. If S-CMA is not relevant in real-world samples, it is less necessary to develop a method to distinguish R-CMA and S-CMA. This study would be more comprehensive if samples containing S-CMA were measured.
3. The introduction is relatively long. Some part of it (for example, the CMA pathway) can probably be moved into the discussion
4. In the Materials and Methods, MS conditions were missing. At least, please add the following information: MS mode (positive/negative), ion source type (ESI or APCI), Block temp., DL temp., CID gas, Nebulizing gas flow, Drying gas flow, Ion spray voltage, CID voltages, and MRM Method (precursor ion -> fragment) for both analytes and IS.
5. How was the LC method optimized? The LC run is long (> 74 min). Have other LC conditions been tried to shorten the run by using a shorter column, a higher flow rate, or a different type of column?
6. Were the MS/MS spectra collected? If so, were they the same for R-CMA and S-CMA after derivatization? It would be better to set up at least one additional MRM to confirm the analytes' identity and improve the method's specificity.
7. Figure 1: It is probably better to move Figure 1. (b) into figure 2. In addition, where is the chromatogram of the internal standard?
8. Figure 2: on (b), (d), (f): a small peak can be found ~ 76 min; is it S-CMA?
9. Figure 3 overlaps with Table 2. Please remove one of them. In addition, it would be better if statistical analysis (such as ANOVA) was used to show significance here.
10. Table 1: only one digital for the slope of the standard curve. Please add more significant figures. In addition, the variation of the recovery should be listed.
Reviewer 2 Report
The article reports the analytical development by an LC-MS/MS technique capable of separating the enantiomers of CMA for determine the absolute configuration of CMA in fruit samples. Accordingly, in this study the authors determined the contents and absolute configurations of the CMA found in three kinds of commercially purchased apple by LC-MS/MS. Furthermore, the presence of optically active CMA in commercial fruit juice was also examined.
The authors should address the following comments:
1) In paragraph 24-27 it is mentioned that: “The structure of CMA (2-methylmalic acid) is similar to that of Mal (Scheme 1).” However, Mal is not shown in this scheme.
2) Also, in scheme 1 (Lines 44-45) there is talk of the formation of a diastereoisomeric mixture of CMA with CIM-C2-NH2, so it would be important to include in the supplementary material the experimental methodology, the 1H- and 13C-NMR spectra and optical rotations of both diastereoisomers to confirm their structure (both for the (R-CMA [15] and S-CMA).
3) In section 3.1. (lines 118-137) the descriptors R- and S-CAM are used, but in the graphs of figure 1 the descriptors D- and L-CMA are used, so then, only use one of the two for more clarity.
Round 2
Reviewer 1 Report
The manuscript has undergone improvements in quality. To enhance it further, the following suggestions should be considered:
1. This study would be more comprehensive if samples containing S-CMA were measured. Especially, in figure 2 (b) (d) (f): a small peak ~ 76 min can be observed. Is it S-CMA?
2. To enhance the specificity of the method, at least one additional MRM (Multiple Reaction Monitoring) should be set up to confirm the identity of the analytes. Specially, in Figure 1 (b), MS/MS of S-CMA derivative has a fragment around 600 m/z which is not present in MS/MS of R-CMA derivative. Can this fragment be used for confirmation MRM?
Author Response
- This study would be more comprehensive if samples containing S-CMA were measured. Especially, in figure 2 (b) (d) (f): a small peak ~ 76 min can be observed. Is it S-CMA?
Response:
Thank you for your indications again. As you suggested, the present LC-MS/MS would be more useful if the presence of S-CMA was demonstrated in the sample tested in the present study.
Unfortunately, we concluded that the peaks in Figure 2 (b), (d), and (f) were not S-CMA, because these small peaks were also detected in the chromatogram of the blank sample, which are samples prepared without apple peel or fruit. Indeed, the peak of R-CMA in fruit of apple was not large as compared with those in peel of apple, and therefore, these noise peaks were relatively stood out. The peak was therefore assumed to be originating from the derivatization reagent CIM-C2-NH2 and not S-CMA.
Thus, following sentences were newly inserted.
Page 5, line 175, > …derivatization reagent, CIM-C2-NH2, because these small peaks were also detected in the chromatogram of a blank sample, which is a sample prepared without apple peel or fruit. The amount present per 100 g of each…
- To enhance the specificity of the method, at least one additional MRM (Multiple Reaction Monitoring) should be set up to confirm the identity of the analytes. Specially, in Figure 1 (b), MS/MS of S-CMA derivative has a fragment around 600 m/z which is not present in MS/MS of R-CMA derivative. Can this fragment be used for confirmation MRM?
Response:
The previous Figure 1 (b), MS/MS spectrum of S-CMA derivative included noise ion peak at m/z approx.600, which confused you. The fragment ion was not used for the confirmation of S-CMA derivative. Therefore, Figure 1 (b) was replaced to another MS/MS spectrum of S-CMA derivative, without the noise ion peak at m/z approx. 600.